# Perceptions of mental health services among the children who are in conflict with the law in Jammu and Kashmir

Mohd Manshoor Ahmed[1] ⓘD and Jilly John[2] ⓘD

[1]Research Scholar, Department of Social Work, Central University of Kerala, Periye, Kerala and [2]Assistant Professor, Department of Social Work, Central University of Kerala, Periye, Kerala

mental health; perception; children; conflict; post traumatic stress disorder

**Corresponding author:**
Mohd Manshoor Ahmed;
Email: ahmed.ssw072106@cukerala.ac.in

## Abstract

**Background:** Due to the Jammu and Kashmir conflict, many teenagers are involved in disputes with the law. The conflict made generations suffer for decades. Such children made the mobs; being involved in life-threatening situations and the risk they confront develop psychiatric disorders. As a result of the various tense conditions when applied in multiple anti-social activities, aberrant children sent to correctional homes have to encounter numerous psychological disorders.
**Aim:** The motive of the study is to explore the level of awareness, availability of services, stigma and obstacles to seeking assistance.
**Method:** Due to the open-ended interview questions and a small sample size of 15 respondents, this study employed a qualitative methodology – a thematic analysis was done.
**Results:** The findings revealed that, although the stigma is not publicly acknowledged, children who break the law and seek mental health services (MHS) are stigmatised. It was also shown that minor offenders fear that when they receive services provided by the staff of the observation home (OH), there will be a violation of their privacy and fear unforeseen repercussions.
**Conclusion:** Collaborative action must proactively raise appropriate awareness to lessen the stigma linked with mental health problems, especially regarding MHS among these teenagers.

## Impact statement

Due to the direct or indirect effects of the surroundings, delinquents are globally impacted by mental health difficulties. One global trend is increased aberrant behaviour caused by conflicts and other situations. As a result, mental health conditions, including posttraumatic stress disorder (PTSD), anxiety and depression, are increasingly prevalent. There are considerable long-term effects of poor mental health on deviant youngsters – those confined to institutions for rehabilitation and mental health needs. There is a significant difference between delinquency and the availability of mental health services (MHS). The vast global mental health treatment gap may be effectively addressed by integrating MHS into correctional settings. Offenders often get assistance from these activities and may be able to attend institutional and community-based therapy based on cognitive behaviour modification. Raising awareness and implementing community-based solutions may significantly lessen stigma, a lack of trust and the fear of unfavourable results. Governments, NGOs and other institutions can use individual, group and other platforms to satisfy juvenile offenders' mental health needs. Notwithstanding the likelihood that these actions could benefit children, caution should be used when putting them into practice since they have not been tried in the real world.

## Introduction

The statement is well known across the globe that Kashmir is a paradise on the earth due to its exceptional natural beauty, climate and geographical location (Malik et al., 2015). J&K was a princely state (Hussain, 2021). Maharaja Hari Singh, the princely ruler, was granted permission to join India and Pakistan as independent nations in 1947. He was first apprehensive about joining India (Mohan, 1992; Farrell, 2002). He put off making a decision on Britain leaving the Indian subcontinent. However, he could not defend his princely kingdom from foreign invasion when the tribes from Pakistan's northwestern areas attacked J&K. He ultimately asked the Indian government to provide the requisite military help (Jamwal, 1998; Singh, 2017; Snedden, 2021). India first ensured that until and unless the princely state formally acceded to India, it would be impossible to provide any military assistance (Abdullah, 1964; Nawaz, 2008). Under this condition, Maharaja Hari Singh signed the Instrument of Accession on 26 October 1947 (Rasool, 2014; Nath, 2016; Paswal et al., 2020). India offered its armed support, and two-thirds of the princely J&K states were

liberated from the tribals of Pakistan (Mangrio, 2012; Kiss, 2013). These disputes led to the country's 1965, 1971 and 1999 wars (Iqbal and Hussain, 2018). After that, in 1987, one political party was not happy with the election results, which produced a sense of resistance among the population, which termed into an armed struggle supported by Pakistan (Parlow, 2012; Majid, 2018). Children in that armed conflict experienced both direct and indirect consequences of violence, such as unlawful recruitment into militant groups (Shah, 2020), killings, gender-based violence (Kumar, 2016), illegal detentions (Ghosh, 2020) and indirect impacts such as lack of access to essential services (Dhamija, 2017). The trend remains, but after the 2008 agitation, the patterns of stone pelting increased (Guroo et al., 2018). The period between 2008 and 2016 was the victim of intensive protests in Kashmir; especially these protests were started from Amarnath Land Row, and a large number of protesters involved in street protests and stone pelting (Tremblay, 2009; Parthasarathy, 2020), and Burhan Wani militant commander was killed in 2016, which also led to severe agitation (Dhamija, 2017). The role of the teenagers in the protest was vital due to the lack of opportunities, employment and critical economic conditions. In other ways, we can say it was the way to diffuse their frustration (Akmali, 2022). However, during the stone pelting and protests, physical injuries from bullets and tear gas shells peaked (Wei et al., 2022). On the other hand, the physical damages were recoverable, but psychological complications were also there, like fear of arrest, night raids and torture. To generate psychological fear among the teenagers, the security forces opted for new ways; they arrested the children who were part of anti-government activities, detained them and kept them in the side army camp, where they would beat them, and their cries could be spread around through loudspeakers to create ear among the youth (*The Wire Staff*, 2019; Haq, 2020). The study stated that the continuity of this violent cycle creates mental health disorders that can lead to depression, PTS, anxiety distress and other mental health complications. Direct episodes of torture may lead to persistent depressive and traumatic conditions (Hassan and Shafi, 2013). Another study's most recent findings revealed that police officers, local health professionals, social workers and other NGOs who offer MHS are seeing an increase in the cases of various psychiatric disorders. The report also mentions the Ganderbal district, where more than 80% of cases were mental health-related, and 80% of patients had psychiatric disorders or had indicators (Syed and Khan, 2017). In Kashmir, depression (41%), anxiety (28%) and PTSD (19%) are quite common (Muntazar: Kashmir Mental Health Survey Report 2015). Nearly 99.7% of the youth have

exposure to conflict, 95.5% experienced psychological trauma, 60% suffer from somatic issues and 91% seem to suffer psychiatric problems. There is a high prevalence of PTSD (49%) among children, especially in areas where the intensity of conflict is very high, and this is due to regular encounters and strong resistance from civilians (Bhat and Imtiaz, 2017). A critical African-based study shows that there are many factors behind getting good services, but there exists deficiency both of awareness and social stigma linked to trust. Still, health service seekers negatively perceive them (Rose et al., 2022).

### Killings and mental health

The above bar chart shows that children experience mental distress when a loved one is killed. However, it has been observed that teenagers in Kashmir experienced mental health concerns during the 1990s owing to killings. The killing of friends during an encounter has been related to psychological issues among young people. The killings were higher between 1990 and 2007 than between 2008 and 2016 Figure 1. As a result, psychological issues were more common during both periods because the cycle of conflict continued, but between 2008 and 2016, injuries were high (Dar and Deb, 2021).

### Injuries and mental health

On the other hand, it has been observed that in the Kashmir region, the injured youth constantly lived in fear of being arrested; therefore, out of that worry, they choose not to go to a hospital for treatment. After 2008, the tendency was anticipated. As a result, many young people were experiencing mental health problems (Amin and Khan, 2009). It has been observed that young people continue to experience phobias (Dar & Deb, 2021). It was discovered that the injuries that occurred between 2008 and 2016 were specifically related to psychological conditions. In Kashmir, persons with severe injuries are more likely to have mental illnesses (Hussain et al., 2017; Dar and Deb, 2021). Figure 2.

### Physical, mental torture, sexual and verbal torture and conflict

Conflict in Kashmir and torture are inextricably linked (Barad., 2020). Physical, mental, sexual and verbal torture always remain unnoticed (Qutab, 2012), but it peaked in 1990 as per data in the bar

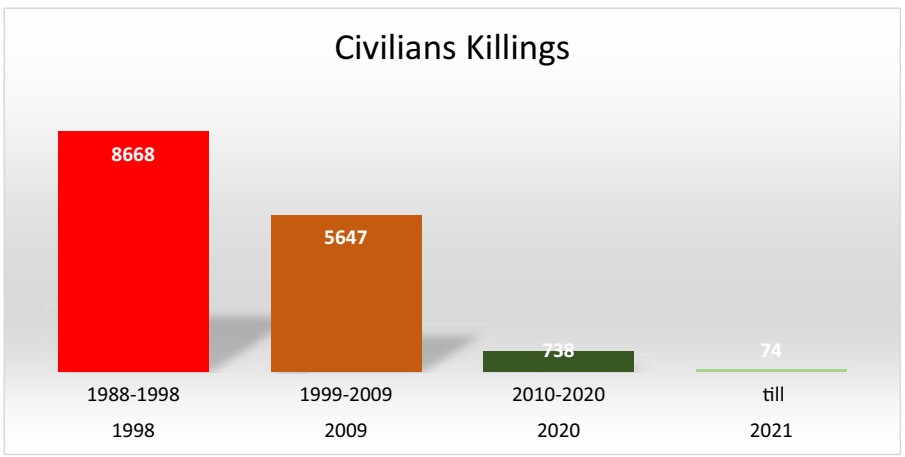

**Figure 1.** Civilian killings year wise in armed conflict. JKCCS, APDP and South Asia Terrorism Portal and Annual Report 2004–2005 (MHA-GOI).

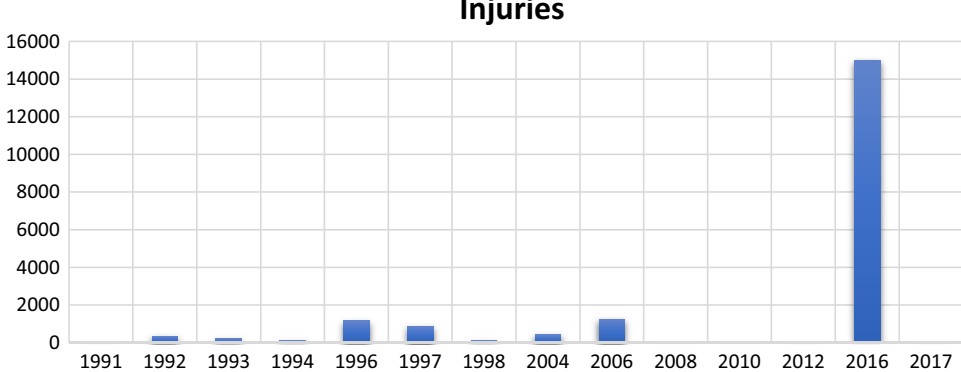

**Figure 2.** Injuries. JKCCS, APDP and South Asia Terrorism Portal.

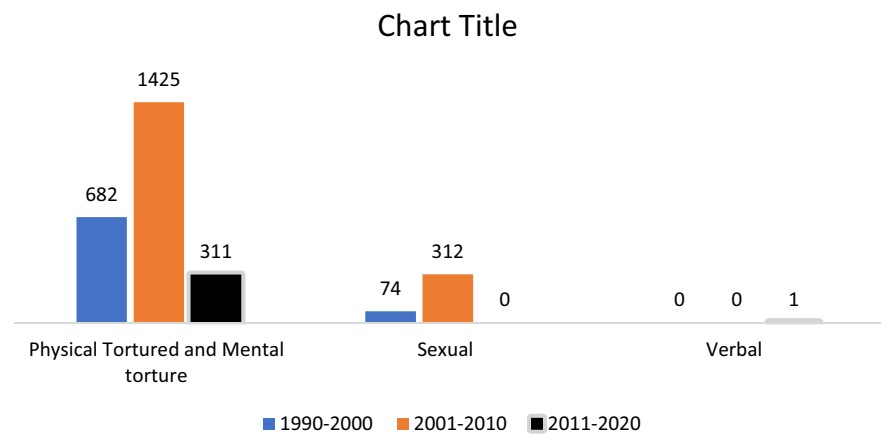

**Figure 3.** Physical, mental torture, sexual and verbal torture. JKCCS, APDP and South Asia Terrorism Portal.

graph (Petersen and Wandall, 1995). However, the trend declined in 2017 in Kashmir (Lalwani and Gayner, 2020). Figure 3.

Overall, mental health has been equally impacted by all factors. However, during the 1990s insurgency's early stages, the general worry of the community was felt, and direct mental illness later began to afflict people (Naik, 2016). Until 2000, killings enhanced mental health problems among the population (Dar and Deb, 2021). Even still, they were frequently injured during the subsequent street protests, and the killing of local militants contributed to widespread emotional damage among the population (Ul Hassan et al., 2017). Additionally, between the 2008 and 2016 protests, torture is not publicly acknowledged (Naik, 2015; Deol and Ganai, 2018).

*The existing system of child protection*
J&K Juvenile Justice (Care and Protection of Children) Act 2013. The J&K Juvenile Justice (Care) Act (J&K JJ Act) is an act that the state passed to codify and amend the law relating to juveniles in conflict with the law, to children in need of care and protection and to juveniles in need of proper care, protection and treatment to meet their developmental needs, for the adjudication and disposition of matters in the best interest of the children and for their utmost restoration (J&K JJ Act 2013). According to the J&K JJ Act 2013, children below 18 years of age have been involved in many offences (Feierman and Shah, 2007). There is still a chance that minors who committed less severe crimes might be apprehended, and the police have the authority to determine whether doing so would be in the juvenile's best interest (Qadir, 2022). The

JJ Act is poorly recognised or adequately implemented, leading to many juveniles engaging in activities that violate the law (Shah and Policy, 2019). While the J&K administration has introduced and passed punitive rules like the Armed Forces Special Powers Act (AFSPA) and Public Safety Act (PSA), it has little regard for juvenile justice or legal protection of minors. The security services flout all juvenile justice norms while unlawfully detaining, assaulting and benefitting from these despicable actions (Gupta, 2021). Children are implicated in an offence, and police abduct them rather than turning them over to Juvenile Justice Board (JJB) and giving custody to a stranger, putting the child at risk of being tortured, sexually abused and imprisoned (Qadir, 2021). On the other hand, parents of minors have limited access to meet. In case of an emergency, they can meet. In addition, the mobility of the juveniles is restricted only within the OH; the official always keeps them watching.

Implementation. J&K have limited correctional homes, one each in Srinagar and Jammu (Ministry of Women and Child Development, 2019). However, both correctional houses need proper infrastructure suitable for such children as counselling facilities. In addition, the staff lack skills and training (Bhat and Mir, 2016; Shah, 2020). A thorough execution of the J&K JJ Act 2013 and the Regulations of 2014 was sought in the petition *Tanvi Ahuja v. State of J&K*. The PIL stated that the state had not established a Child Welfare Committee (CWC) or JJB. The case determined CWC and JJB in the state in response to the petition. The state had never

followed the JJ Act of 2015 requirements. India abolished its prior juvenile law to implement the requirements of the UNCRC into its national legislation and to protect children's rights (Bajpai, 2018). It enacted the Juvenile Justice (Care and Protection of Children) Act in 2000 (Mehta, 2008). In addition to offering a regulatory structure for adolescents' safeguarding, management and reintegration to meet their cognitive concerns, this Act calls for an exceptional approach to avoiding and curing delinquent behaviour. In deciding cases involving youth violence, it adheres to the maxim of what is in the children's best interests. The juvenile law was passed in India, but because of its unique legal connection, it did not apply to J&K (Muncie, 2008). Though it took a while, the J&K eventually passed a juvenile justice law distinctive from the JJA in India. In 1997, the state of J&K passed the JJA of India from the previous year, and it has been effective till 2013. The J&K approved the JJA that India enacted in 2000 in 2013. This demonstrates how seriously J&K takes adolescents (Shah, 2020; Jasrotia and Sunandini, 2022).

**Individual care plan.** In the J&K JJ Act of 2013, there is no mention of an individual care plan, while the JJ Act of India 2015 discussed the individual care plan. The J&K JJ Act requires such certification of establishments working with children who need care or protection (J&K JJ Act of 2013). The JJ Act of 2015 emphasised individual care plans and required registration of all childcare facilities (JJ Act 2015). The national government abrogated Article 370 on 5 August 2019 (Chandrachud, 2019), and the Juvenile Justice (Care and Protection of Children) Act 2015 became applicable to J&K in 2021. There is optimism that minors will get more significant advantages which were not enlisted in the J&K JJ Act 2013. That national law will ensure that the individual care plan holistically makes child growth possible and teaches about the child's past. Through direct child participation, they can discuss future objectives, emotional well-being, individual health, academic and entertainment needs. A thorough plan encompasses every person's developmental, psychological and intellectual development (Shah, 2020; Jasrotia and Sunandini, 2022).

**Government initiatives for the youth.** The individual care plan can include the children's future goals and preferences. The Government of India has also developed numerous rehabilitation programmes for these youths (Aslam and Sudan, 2021). Additionally, the army has launched multiple missions for Kashmiri youth, like Sahi Rasta initiative (Zade, 2017; Dongare, 2022). The political process in J&K is inactive after the revocation of Article 370 (Khan, 2021) because the LG Administration, Government of India, directly manages and oversees the entire process (Warikoo, 2010).

*Theoretical framework (the ecological model)*
When examining the perception of MHS among minor offenders, it is crucial to consider their difficult living conditions, family history, support and neglect. These factors contribute to youngsters breaking the law and acting disrespectfully towards others (Repetti et al., 2002).

**Microsystem.** A pattern of actions, societal roles and interpersonal communication often occurs in the direct vicinity where the individual is enmeshed, referred to as the microsystem (Bronfenbrenner, 1994). Frequent encounters between juvenile offenders, individuals and groups within society are closely related to their misbehaviour. The next part will focus on four elements closest to the particular teenager at the microsystem level: youth traits, family connections, friendship affiliations and school attachment. Adolescent features, including socio-demographic characteristics (such as gender and

age) and psychological factors (such as the capacity to exercise self-control), have been demonstrated to influence the likelihood of delinquency (Vanniasinkam, 2010; Agha et al., 2012; Khan, 2021).

**Gender.** Most academic research in Kashmir, Palestine and Syria has proven that male adolescents are substantially more likely than females to engage in deviant acts (Farrington et al., 2016; Slone et al., 2017; Bhat, 2019). There are apparent differences between male and female teenagers in Kashmir (Parashar, 2009; Shekhawat, 2014). Males were shown to be more prone than females to participate in stone pelting, natural aggressiveness, self-centred behaviour and aberrant behaviour, according to the study's findings, and also likely to hold deviant opinions and have unsavoury associates (Adenwalla, 2006; Shah et al., 2019; Ganie, 2021; Hassan, 2022). Compared to their female counterparts, male adolescents were more traditional and likelier to engage in aberrant behaviour (Roy, 2008; Clarke, 2011).

**Age.** Several research conducted in Kashmir has shown positive results on the age influence on teenage delinquent behaviour and mental health (Anjum and Varma, 2010; Bhat et al., 2017; Hassan, 2021). On the other hand, studies have observed a significant positive relationship between age and frequency of deviance. Western studies have proposed that the prevalence of juvenile offending follows an age–crime curve that rises rapidly during early adolescence, peaks in mid-late teenage years and steadily declines after that period (Grasmick et al., 1993; Loeber et al., 2012; Gottfredson and Hirschi, 2022). According to most research, there is a clear correlation between early adolescence and increased aberrant behaviour and middle youth; however, this relationship weakens as children become young adolescents (Farrington, 1994; Farrington et al., 2009). However, in the context of Kashmir, similar trends have been observed (Elbakidze and Jin, 2015).

**Self-control.** According to J&K-based studies, self-control is a potent and accurate predictor of deviant and mental health issues (Ashiq, 2015; Iqbal et al., 2020). Self-control is a steady interpersonal quality that may significantly reduce people's urge to engage in some abnormal behaviour (Gottfredson et al., 1990). Research on 1,500 children has indicated that those who lack self-control are more likely to engage in offensive activities (Cutrín et al., 2015). Moreover, other abnormal behaviours like snatching weapons (Kaura, 2017) and involvement in stone protests are a result of poor self-control (Kak, 2010). Also, among youths in rural regions, a lack of self-control is strongly connected with interpersonal physiological aggressiveness (Var et al., 2011; Fayaz, 2019; Ganie, 2023). Low self-control linked to gang membership, violent behaviour (Naz et al., 2016), substance use, theft and crime among Jammu youth has been observed (Sharma et al., 2022).

**Relationship with parents.** Family had a significant influence on the lives of teenagers in J&K (Munshi et al., 2008). Family history, support and neglect are crucial for their difficult living conditions (Repetti et al., 2002). A prolonged conflict between mother and father impacts children's mental health for a long time because they have to stay with them for a long time; such action may affect them and cause developmental disorders (Vahedi et al., 2019). A weaker bond between parents and children is associated with a greater propensity for delinquent behaviour (Hussain & Imtiyaz, 2016; Hussain, 2014). Another gender-based study on 640 Palestinian adolescents shows similar findings (Punamäki et al., 2011). Children who are not closely watched have a greater incidence of protest participation (Hussain et al., 2017). Children's drive for independence and self-identity has been proven to make the controlling

parenting style a source of conflict and miscommunication (Ayub et al., 2012). A study has examined how various family factors affected juvenile misbehaviour in J&K. Another study has indicated that teenagers of unemployed parents (Majeed et al., 2020) display much greater rates of delinquency and violence because their parents lack parenting skills (Akhter et al., 2011). Similar findings affirm the concerns of an ineffective parenting style in J&K (Zhang et al., 1995; Amin, 2011). Also, research has shown that difficult home situations with components like forceful parenthood, heated disagreements and parental abuse are favourable conditions for disruptive and delinquent behaviour (Fagan, 1995; Nisar et al., 2015; Hussain et al., 2017).

Peer effect. In Kashmir, having delinquent friends is especially notable as a determinant of delinquency (Tremblay et al., 1995). The loss of parental authority is a sign of shifting social patterns in teenagers' life, from family bonds to peer interactions, due to ongoing conflict and frequent changes in recent decades (Malik and Bhat, 2022). Peer groups are a significant source and reinforcer of violent behaviour in adolescents (Ingram et al., 2007b, 2011; Ferguson and Meehan, 2011). Another research has shown that teenagers who view their closest friends as deviant are likelier to participate in delinquency (Rokven et al., 2017), increasing their propensity to protest, stone pelting in Kashmir and delinquency in Jammu (Mathur, 2016; Khanna et al., 2020). Neighbourhood, peers and exposure to various conflict-related factors affect children's mental health (Jaggers et al., 2016). They are owing to low academic performance and a lack of rapport between teachers and adolescents (Suri, 2014; Malla, 2019; Dar & Deb, 2020).

Mesosystem. The growing human is contained inside each of two or more microsystems that make up the mesosystem (Bronfenbrenner, 1994). Adolescents' peer interactions and academic achievement may impact their parental relationship experiences. These connections can help explain how delinquent behaviour works. Researchers have noted that teenagers with poor parental connection and limited self-control are considerably more prone to use aberrant coping mechanisms when faced with stressful situations (DuBois et al., 1994; Laursen and Collins, 2009; Ardelt, 2010). On the other hand, adolescents who have disruptive peers exhibit more delinquent behaviour and less self-control (Hussain, 2012; Sharma and Marimuthu, 2014; Azim, 2019).

Exosystem. The exosystem, depicted as a surrounding scope encompassing the interconnections between various engagements or circumstances, is where the developing person is only actively engaged in one encounter or circumstance (Bronfenbrenner, 1994).

Socioeconomic conditions. Socioeconomic disadvantage potentially contributes to increased delinquent involvement (Anjum & Verma, 2010; Rashid and Waddell, 2019). A Kashmir-based study shows that economic constraints are significantly associated with minors' substance abuse and delinquent behaviour (Dabla, 2012; Vishwakarma, 2021). Findings depict that the absence of bread earners is significantly associated with delinquency and mental health problems (Sameena et al., 2016); parents going to work can lead to inadequate supervision of their children, contributing to the development of delinquent behaviour among the minors (Flanagan et al., 2019). Studies on adolescents confirm that family economic condition is directly proportional to the prevalence of delinquency in J&K (Van der Westhuizen and Swart, 2015; Bhorat et al., 2017). Several studies in Kashmir demonstrate

that young people participate in numerous protests and terrorist activities due to their poor socioeconomic condition (Anjum & Verma, 2010; Narain, 2016; Singh, 2019).

Community. Children are embedded in the community; underprivileged circumstances might encourage delinquent behaviour and affect children's mental health (Garbarino et al., 1991; Boyd et al., 2022). In Kashmir, due to community-based violence and street protests (Ganie, 2023), another study has proved that contact with extremist groups, fear of raids and arrests foster more excellent deviant behaviours among minors (Dar et al., 2020). Exposure to poor conditions like frequent protests, the impact of neighbourhood, encounters and participation in militants' funerals are more likely to evoke psychological stress in adolescents, which in turn contributes to an increased likelihood of deviant behaviour in Kashmiri youth (Mushtaq, 2012; Dhamija, 2017; Malik, 2018). Neighbourhood impacts more; youths spend the most time within the community (Durlak and DuPre, 2008). What kind of attitude does the community have? In other words, we can say that they are very much familiar with the environment of their community (Brooks-Gunn et al., 1993). So, it is easy for them to abstract the particular condition. They can act comfortably (Johnson, 2005). In 2016 there were 65 gun-snatching incidents (Mohanty, 2018).

Mental Health facilities. The studies on the mental health of children who conflict with the law need to be more concentrated (Teplin et al., 2002). There is, however, limited research on the treatment of young patients with psychiatric issues (Mushtaq and Fatima, 2016). Multiple factors are responsible for the cause of various mental health disorders among children who conflict with the law; they require mental health attention, or they need MHS (Almanzar et al., 2015) as it is well understood that mental health of the general population of Kashmir started to suffer after 1987 when resistance started against the government. Children, elders and women have experienced many traumatic incidents during the insurgency (Naik, 2016). To ensure that suitable therapies are offered to such children (Henggeler et al., 1997), it is crucial to understand how juvenile offenders feel about the effectiveness of mental healthcare. OH officials offer mental health assistance to detained youngsters after bail or during detention (Swank and Gagnon, 2016). However, evidence indicates that these services' efficacy is dubious (Malla et al., 2019). The inadequacy of services provided while juvenile delinquents are in OH is simply a forerunner to the ineffectiveness of various interventions once they return home (Zeola et al., 2017; Shah, 2020). The strategies to cope with such kinds of mental disorders commonly adopted by the people in Kashmir are spiritual practices and other community-based approaches. In other words, not only the elders or adults are entitled to such techniques. The involvement of teenagers is also expected (Aqeel et al. 2017). This division of responsibilities between health and rehabilitative services, as well as the absence of intersectoral coordination in providing these services, can be associated with adverse effects for the discrepancy in the delivery of health and rehabilitation aids (Storm et al., 2020).

Macrosystem. The macrosystem level considers elements of the larger environment, including cultural values, way of life and opportunity structures that eventually impact the social structures and activities at the immediate system level (Bronfenbrenner, 1994). In particular, where Kashmiri culture and religion are strong

influences, boys and girls are treated differently throughout establishing gender roles (Parlow, 2011; Behera, 2016; Gabel et al., 2022; Kazi, 2022; Nisar, 2023). Kashmir has a substantial gun culture which immediately affects the young (Tremblay, 1996; Parashar, 2011). Many teenagers under the influence join various militant groups (Hilali, 1999; Oberoi, 2004; Ahmad and Hussain, 2011). However, when fighting with forces, many young individuals perish (Ahmad and Balamurgan, 2020). Due to strong religious feelings, many will refer to it as a sacrifice (Devadas, 2018). Often different militant organisations publicly exhibit their weapons (Whitehead, 2022), directly affecting young people. Children can view it as a distinctive aspect of the culture. Their behaviour may be abnormal due to that culture (Jamwal, 2003; Dorjay, 2016; Shah and Policy, 2019). According to numerous studies, there is a significant chance that children from families where delinquency has a history may grow up to be classified as delinquents (Egalite, 2016; Kaiper-Marquez et al., 2021). Intense community shaming and reconciliation will likely cause labelled persons to feel humiliated once they are reintegrated into the community in such a collective society, which may drastically lower their identity. Such minors are more prone to deviance and susceptible to mental health concerns (Sharpe, 2015; Stuewig et al., 2015; Lageson, 2016; Schalkwijk et al., 2016).

### Mental health services, stigma and trust

Ample studies discuss the relationship between stigma and MHS among teenagers. Stigma is linked with the mentally ill, which could harm their growth, development, self-esteem, daily activities and seeking appropriate treatment (Hinshaw, 2005; Kaushik et al., 2016). Trust in counsellors or those who assist MHS is widely ignored in the stigma studies (Gilburt et al., 2008). The study of MHS has also paid little attention to trust (Brown et al., 2009; Gaebel et al., 2014). However, if we talk about interpersonal relations between the service seeker and the service provider, trust plays a vital role (Terrell and Terrell, 1984; Bucci et al., 2016). Because it is a matter between two individuals primarily, on the other hand, stigma may be from the social side as well as self-stigma. In addition, trust remains ignored in the MHS domain (Gilburt et al., 2008). This is unexpected because the interaction between service providers and clients of MHS is thought to be the most crucial process (Brown and Calnan, 2013). A psychologist has established a concept called trust level theory. The efficiency of the systems as a result of how the processes interact is said to be determined mainly by trust level. According to Gibb, trust is an innate, unplanned emotion similar to passion (Gibb, 1997). However, factors include the trustworthiness of the healthcare providers' patient-centeredness, the assurance that policies will not negatively impact individuals, the healthcare professionals' skill, the care benchmark and the flexibility and coexistence of the services concerning spreading knowledge, accessibility, privacy and possible vulnerability (Blomqvist, 1997; Straten et al., 2002; Thompson et al., 2003; Shou et al., 2011). The majority stress the optimistic acceptance of a vulnerable situation in which the truster believes the trustee will care for the truster's interests (Hall et al., 2001, p. 615). A crucial component of trust is the belief in loyalty, competence, fairness, privacy and general reliance (Hall et al., 2001). On the other hand, we will talk about stigma and MHS; stigma is a significant obstacle behind MHS (self and public stigma) and enhancing mental health problems (Polaha et al., 2015). The study shows that children who conflict with the law experience stigma and a determined-to-trust deficit, making their mental health more pathetic (Naik, 2016; Mir and Bueno de Mesquita,

2022). Though stigma is thought to prevent mental healthcare participants from rehabilitating, the exact intermediary procedures within MHS providers are still unknown (Mezey et al., 2016). As engagement among individuals and professionals is a core component within MHS and trust is viewed as a central aspect of this communication, trust may be considered a relatively essential intermediate factor connecting stigma to the effects of MHS (Walsh et al., 2011).

### Conceptual framework

Individual-level variables cannot fully explain adolescent behaviour (Loeber and Farrington, 2000; Anderson et al., 2015) because communities defined by geography (Shannon et al., 2021) and conflict (Yule et al., 2003; Slone and Shoshani, 2022), and groups determined by gender, school and income, appear to exhibit consistent, non-random patterns of teenage problem behaviours (Loeber and Farrington, 2000; Anderson et al., 2015), parent's profession in the earlier research (Chang et al., 2016). Several studies have shown that a child's environment may function as a potential cause or a shield against delinquency (Ferrell, 1997; Nation et al., 2003), mental health (Betancourt and Khan, 2008; Williams and Thompson, 2011) and behaviour issues (Margoob et al., 2006; Thabet et al., 2006; Ayub et al., 2012). Consequently, we constructed an integrated model that accounts for the influences of microsystems, mesosystems, exosystems and macrosystems on adolescents. They may experience various psychological issues due to the environment, which may indirectly or directly contribute to the development of deviant behaviour (Lambie and Randell, 2013). Afterwards, these minors moved to correctional settings. They could be hesitant to use mental health treatments because of stigma and mistrust, as depicted in Figure 4. We came up with hypotheses regarding the stigma associated with MHS. In particular, the latter model hypothesises that aberrant youths need more faith in guidance counsellors and, despite this, in OH staff members. Children reportedly feared confidentiality would be violated since the visiting counsellor might tell JJB or other officials everything.

### Objectives and rationale of the study
#### Objectives.

1. To learn about the socioeconomic situation of the children who are in conflict with the law in J&K.
2. To understand the stigma associated with children who are in conflict with law and are housed in correctional settings.
3. To understand the fear of unforeseen repercussions among juvenile delinquents in J&K.
4. To understand the trust in service providers among the children who are in conflict with the law.

**Rationale of the study.** The rationale behind choosing this particular area is because it has yet to be given much attention (Noor and Llah, 2015; Bhat et al., 2021). However, the prevalence of mental disorders among delinquents is high in Kashmir (Hassan, 2021). Nonetheless, Kashmir has a high rate of mental illness among offenders (Goldstein et al., 2005; Housen et al., 2017; Paul and Khan, 2019). The researcher has discovered through literature review that specific issues of young people who were in trouble with the law and had spent time in correctional homes in J&K are not extensively documented. The lack of study and effective intervention in this area leaves juvenile mental health issues exposed and stigmatised for a very long period. It should be mentioned that the social welfare authorities, academic planners, healthcare authorities

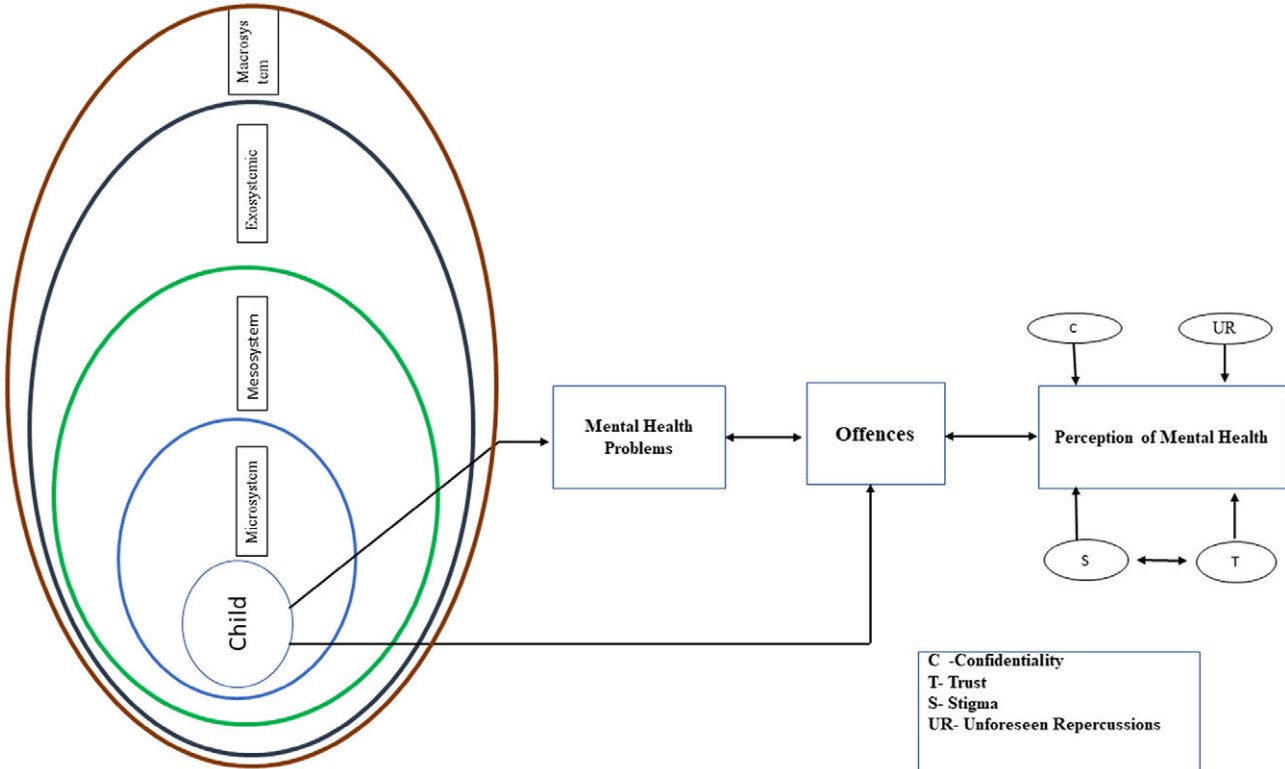

**Figure 4.** Conceptual framework. The framework aims to explain how ecosystems can affect children from a micro to a macro level since they interact directly and indirectly. Suppose an issue arises, for instance, at the macro or family level, both circumstances have the potential to influence how minors behave. As a result, individuals can develop mental health issues or act inappropriately.

and social workers are unaware of the psychological concerns that offensive children experience due to a lack of appropriate and reasonable engagement with the issues. As a result, individuals are unable to manage their problems consistently. This study recognises this as a significant societal issue in J&K and tries to provide practical, participant-friendly solutions for the target group in correctional settings. This proposition would benefit policy-makers, government representatives, ICPS (Mission Vatsalya), social welfare administrators, psychologists and social workers to put new guidelines and plans into action for the welfare of delinquents.

### Operational definitions of anti-social activities
1. In this study, anti-social activity is a disturbance in the social order through noisy protests and disruption of day-to-day activities. Protest leads to closing all business activities (Bhat, 2019; Hassan, 2022).
2. Anti-social activities are actions taken by an individual or a group that violate social laws and ethical standards.
3. The term "anti-social activities" describes a collection of actions that may undermine the regular function of life and affect essential community service delivery.

### Method
**Study design.** The study aimed to examine adolescents' perceptions of mental healthcare. Due to interviews producing open-ended replies, this research used a qualitative methodology, and the sample size was restricted to 15 individuals. The study permitted a comprehensive face-to-face examination and analysis. There was no defined standard instrument. Thus, questions were focused on the literature's primary topics. The researchers expected that

participants' replies would indicate that using MHS carries a stigma, such as being seen as weak by peers. Moreover, authorities may see these youngsters as weak. It was also suggested that delinquents might fear unforeseen repercussions while reporting the need for mental health assistance, such as losing their candidacy for school enrolment or missing employment opportunities. These adolescents may also be seen as fragile by both authorities and peers. In addition, the study hypothesised that youngsters in legal peril do not trust counsellors or OH authorities. As the counsellor is also accountable to the JJB, there is a concern that confidentiality may be compromised.

**Sampling.** Srinagar and Jammu districts have the highest number of delinquents in J&K (Hassan, 2022; Jasrotia and Sunandini, 2022). However, 15 youngsters were picked using the snowball sampling method from the districts of Srinagar (eight) in the Kashmir division and Jammu (seven) in the Jammu division. The participants were between 10 and 18 years old. The adolescent must have spent at least three months in a correctional facility, and all participants were bailed out.

**Data collection.** Before the interview, participants' demographic information was gathered via a questionnaire. The demographic questionnaire covered age, educational level, family profession, family status (APL or BPL), parent education and participation in delinquent behaviours. All demographic questionnaire items were close-ended, and participants had 10 min to respond. The questions for the interview were open-ended. In addition, it was anticipated that this study would inspire more research into establishing guidelines for such juvenile delinquents. Before conducting the interviews, the researchers provided a debriefing to describe the purpose

**Table 1.** Demographic profile of the respondents

| S. No. | Gender | Age | Crime | Grade/class | Religion | Area |
|--------|--------|-----|-------|-------------|----------|------|
| 1 | M | 15 | Stone pelting | 9th | Muslim | Kashmir |
| 2 | M | 14 | Stone pelting | 8th | Muslim | Kashmir |
| 3 | M | 17 | Stone pelting | 11th | Muslim | Kashmir |
| 4 | M | 14 | Stone pelting | 8th | Muslim | Kashmir |
| 5 | M | 15 | Stone pelting | 8th | Muslim | Kashmir |
| 6 | M | 14 | Stone pelting | 8th | Muslim | Kashmir |
| 7 | M | 16 | Stone pelting | 10th | Muslim | Kashmir |
| 8 | M | 15 | Stone pelting | 9th | Muslim | Kashmir |
| 9 | M | 16 | Murder | 10th | Hindu | Jammu |
| 10 | M | 17 | Rape | 11th | Muslim | Jammu |
| 11 | M | 16 | Robbery | 10th | Hindu | Jammu |
| 12 | M | 14 | Drugs | 8th | Hindu | Jammu |
| 13 | M | 15 | Drugs | 9th | Hindu | Jammu |
| 14 | M | 16 | Rape | 10th | Hindu | Jammu |
| 15 | M | 15 | Robbery/theft | 9th | Hindu | Jammu |

*Note*: Males aged 10 to 18 were enrolled in the average 8th to 12th grade. Nine respondents identified as Muslims, eight from Srinagar and one from Jammu, while the remaining identified as Hindus from Jammu. Eight individuals who participated in stone pelting were from Srinagar, while others committed offences from Jammu.

of the study and evaluated the eligibility of the children to be interviewed. The aberrant youngsters signed a permission form.

**Procedures.** The participants were individually questioned. Researchers made use of field notes. For the participant, the interview lasted 30 min at maximum. In some cases, it reached 45 min. A separate setting was chosen for the interview to ensure the participants' comfort during the procedure.

**Data analysis.** Thematic analysis, which is open-ended, dynamic and can provide additional insight, led the data analysis to further our knowledge of how children form their opinions of MHS (Bloomberg, 2012). Getting familiar with the participant data involved transcribing, reading and rereading the transcriptions, making notes and creating emergent themes by systematic coding of interesting features across the entire data set, compiling information pertinent to each code, looking for themes by organising codes into potential themes and collecting information pertinent to each prospective theme (Booysen et al., 2021). Ultimately, data analysis and drafting of the findings have been through an iterative procedure, with the authors debating and commenting on the data analysis (Bloomberg, 2012; Gergen and Gergen, 2008).

*Results*
A review of issues found while analysing the data will subsequently happen – exposition of the 15 children involved in aberrant activities like stone pelting, robbery, rape and murder.

**Parents' education.** Regarding parental education, four fathers were educated up to the 10th grade. Six fathers have less than a 10th-grade education; three fathers never went for studies; two parents attended college. Eight of the mothers had education levels below the tenth grade. Seven did not go to school.

**Socioeconomic status.** Few respondents self-identified as below the poverty line (BPL), despite most respondents' socioeconomic

status falling into this category. Teenage delinquency and income seem to be related in some way (Hassan et al., 2020). Notably, just one parent worked in the public sector, and 13 were labourers. The majority of fathers and mothers were lower-class income earners. The majority of lower-income households reported making do with their little incomes. One delinquent was an orphan, whose father had died (Table 1).

**Offence repatriation frequency.** According to the respondents' revealed information on the frequency of offence recurrence, one respondent committed a crime seven times, two minors committed a crime three times and four others engaged in aberrant behaviour four times. Also, five children engaged in offensive conduct twice (Table 2).

*Theme 1: Understanding of mental health*
Sub-theme: Awareness/explanation of mental health from children's point of view. The concept of mental health was seen differently by children. Others characterised mental health differently, dismissing any issues and offering a medical explanation for their symptoms instead.

1. "When someone is disturbed, they are said to be mental."
2. "I believe that before becoming insane [Pagal Panthi], I was quite aware; my brain was blocked."

Sub-theme: Disturbance of mind and rejection of mental health problem. Despite their linguistic barriers, several respondents could link explicitly "disturbed" with their mental health. These statements were fairly lucid and would be understandable. Although some children could express their issues plainly, others did it in a more ethereal manner. This may have resulted from simple language difficulties or represent specific regional viewpoints. They could interpret their experiences and show profound understanding by attributing the situation to the brain. Even when specifically questioned, several children denied having mental

**Table 2.** Themes

| Themes | Sub-themes |
|---|---|
| 1: Understanding of mental health | 1. Awareness/explanation of mental health from children's point of view<br>2. Disturbance in mind<br>3. Rejectioof mental health problem |
| 2: Who was available for MHS | 1. Lack of mental services |
| 3: Being with fellow delinquents | 1. Communicationa<br> a. Negative viewb<br> b. Positive views, talk and sharing |
| 4: MHS covered in screening in JJB/OH | 1. Dealing with psychological problems<br>2. Availability of MHS in locality |
| 5: External pressure | 1. Officials' pressure for seeking the MHS |
| 6: Encouragement of MHS | 1. Encouragement of MHS by officials |
| 7 & 8: Stigma and barriers | 1. Obstacles to MHS<br>2. Fear of unforeseen and trust deficient |
| 9. Sleeplessness and aggressive behaviour | |
| 10: Awareness level (outreach) | 1. Individual engagement sessions<br>2. Community- and official-level engagements |
| 11: Comfortable with MHS | |
| 12: Confidentiality | 1. Fear of privacy breach and breakdown of confidentiality |

health issues These individuals tended to attribute the reason to somatic symptoms.

3 "Respondent claims that even though I have a headache and sore eyes, I am not ill. It's very typical among the others."
4 "I don't suffer from any mental illnesses. I recognise the spooky dream. Sometimes, I skipped sleep."

These responders describe their mental state in various ways. For instance, several respondents blamed their health issues on particular physical rather than disturbance of the mind. As the interviewer pressed for further information, he was sure he had no mental health issues. The responder shared this rejection of mental health issues.

5 "I'm not a mental problem."
6 "I refuse to go to the counsellor to be among the mentally ill children."

Even after admitting mental health issues, individuals continued to have unfavourable opinions on their own and others' psychological health. It is common for young individuals to discuss mental health negatively (Oral et al., 2016).

*Theme 2: Who was available for MHS*
Sub-theme: Lack of MHS. Most of the respondents said no MHS were offered in the OH. Few stated they were told a counsellor was available to talk to them. One said, for example:

The office of the OH suggested that we consult the counsellor if we needed assistance, but I hardly saw any counsellor while in the OH. They mentioned that a counsellor has been on duty, and you might approach them for counsel if necessary. If you notice someone battling, see what you can do to encourage them to seek MHS. I attended a medical camp, but none of the psychiatric came there; I consulted with the psychiatrist with the help of my family. If we complain about our mental health problems, we must follow many steps during the consultation procedure.

Narrative of another teenager:

One of the participants shared that they did not provide direct services, but sometimes the official from the OH came and asked whether they required any help; you can ask us, but there was no explicit mention of MHS. If you are facing any problems, you can share them with us. Most of the time, they conducted medical camps, but no psychiatrist or counsellor. However, no one considered that we were distressed.

Who was accessible to the convicts about the connected MHS was the topic of the associated MHS. After analysis, we can say there was a lack of MHS.

*Theme 3: Being with fellow delinquents*
Sub-theme: Negative view regarding communication. "Why do I have to know anyone? Being alone is better since no one will give you support when things go difficult."

"Since you're not doing it in a group but are going to communicate to one individual alone, one must have communicated with them individually."

In another interesting finding, a few respondents said they never talked to anyone about their offence.

Sub-theme: Positive view regarding communication. "It's a great idea to talk to anyone, as one of the participants said, particularly with family and friends. It may help to resolve a tough problem."

One respondent said that "talking to others is delightful, especially mates".

"When you are there, you interact with your friends who are with you and just your friends. You truly don't want to share your troubles with your family. Talk to everyone there who is experiencing it with you."

Most minors felt more at ease speaking with others who understood their issues or had comparable experiences. Most importantly, most adolescents felt at ease communicating with peers inside the OH compared to the counsellor.

*Theme 4: MHS covered in screening in JJB/OH*
Sub-theme: How to deal with psychological problems. Very few stated that they were briefed about dealing with various psychological stressors while returning from the OH, but not profitably or professionally.

"Counsellor advised to play games, calm anger, talk to parents about difficulties, and not ponder too much before leaving OH."

"They discussed facing your emotions rather than ignoring them, not ignoring problems, and the need to seek counselling or support from a mental health professional if you are struggling."

"When I was sent there, I was anxious and afraid, but no one asked me about my condition."

No professional counselling was given to the minors.

Sub-theme: Availability of MHS in locality. "Amm, I notice that there are no essential health services in my neighbourhood, and I've never heard of any MHS there. No doctor of for mind."

"I never see a doctor for other diseases. To take treatment, our people go to the city. Doctor-related to mind is the secondary option."

The rest said that officials asked that all services be accessible in your localities, and most responded similarly. The comments reveal insufficient screening for mental health problems. So no services were available in the neighbourhood.

### Theme 5: External pressure

The external pressure encompasses staff members' security, influencing juvenile delinquent's mental health and seeking HMS.

### Sub-theme: Officials' pressure for seeking MHS.
Most of the youngsters stated that they did not have such kind of pressure from the side of the officials, and officials were never very serious about MHS. Nearly half of the youngsters stated that the officials should have done something or taken the lead to enhance MHS. The rest of the children told that officials thought they were not serious if they felt they needed any benefits such as MHS.

> One of the children stated that it is the responsibility of the children in the OH to push the officials to promote MHS. It needs an hour because most of us suffer from various kinds of stress and strain. MH is not a non-topic of discussion with authorities. I've never actually talked about it with anyone else. I never really bothered discussing it, or nobody ever brought it up. We hardly seldom get together and discuss officials. Even if no one doesn't care, there are situations when sharing information with them might have unexpected effects. We don't think about that discussion much.

Children were not encouraging MHS after analysing the theme, although they sometimes needed to be urged more. Few believed that occasionally the government promoted MHS, but it was the responsibility of the children to avail them – no direct mention of pressure from the side of the OH staff.

### Theme 6: Encouragement of MHS
### Sub-theme: Encouragement of MHS by officials.
"Particularly, authorities showed little concern about pains and made no effort to MHS. They request to take a pill, leave, and go to bed."

"If we reveal everything, they've been pushing only the other things, ultimately affecting your future. Nobody is discussing mental health."

"Somehow, when I was unhappy and in a bad attitude, authorities recommended that I visit a counsellor."

"So, it's like, not simply to speak to them because they don't, they don't listen as they listen, but they're not hearing since the only thing they are looking to is, I don't know, I honestly find it pointless talking to officials."

The researcher found inconsistent responses from various staff members.

"Many don't care, so yes, that's the case."

Some respondents believed that officials need more awareness of difficulties in OH. As such, they think they need to be better placed to support them adequately.

"These are youths with multiple problems that nobody else is conscious of since they believe they are okay and all that, but which must be dealt with because, you know."

Maybe they would because they have not known, but they would not ask you questions if they see you in a bad state.

"Indeed, because they avoid discussing that particular psychological condition, they speak about how others make fun of them, but there must be other reasons to feel anxious."

Some respondents felt unsupported in coping with issues because they believe others are prejudiced against offending actions.

Most respondents claimed no support for promoting MHS. Children claimed that although there was no promotion of MHS, we could communicate with other teenagers in the OH when things get difficult or stressful. We may enjoy conversing with others in any way. And if we communicate with one another, our chances of finding alleviation from this type of condition improve. However, most teenagers fear sharing their matters with others because they worry about their future, whether they will be accepted at a school or find jobs, and that their privacy would be violated. After analysing the theme, there needed to be more encouragement. Few felt that sometimes officials encourage MHS, but they feel it is up to the children to get them.

### Themes 7 & 8: Stigma and barriers
### Sub-theme: Obstacles to MHS.
Children stated that they have to consult with a psychiatrist or counsellor to cope with a particular situation; with that, some children said no one can understand the actual problem of anyone. They have to deal with it themselves. Few of them were unaware of the obstacles to MHS.

From most respondents' responses, it was found that there was a stigma attached, which was not openly acknowledged since they did not discuss their direct experiences.

For instance, one said: "Do not be encouraged; simply speak to someone if you need help. Do not let it increase. Just discuss it with someone like a friend. Everyone in the room was believed to have psychological issues or abnormalities. Just let someone know about it."

Juveniles do not trust counsellors, so they are also comfortable sharing their problems with mates and unnoticed stigma.

"Because we are supposed to put up everything that goes along with being juvenile offenders, we thought we were weak if we went and spoke to someone about mental health."

**Self-stigmatisation:** Low self-esteem and low confidence may stop them from seeking services.

The theme in this question dealt with stigma. Most of them fear of their career and did not want to see them as weak.

According to one respondent, children are outgoing, sincere and honest. He said, for instance:

> I'm not sure. It doesn't appear to be bad. Our correctional home inmates dealt with things in a rather transparent manner. Just be real and sincere. Now we'll be able to see whether you're stalling. One such man murdered his girlfriend after suffering a total mental collapse. We won't hold him responsible. He's not to blame. Another issue would arise if you claimed to have been traumatised while not seeing anything. No one should encounter any issues as long as you are being true and honest. The only thing I can think of is that.

### Sub-theme: Fear of unforeseen repercussions and trust deficient.

### Fear

Due to responses that have led them to lose significant associations like close friends or even partners, most respondents preferred to refuse having mental illnesses. This behaviour may be prompted by fear of the condition.

> I will tell you about my experience in the observation home. Several staff members tell us that we have nothing, and the people trying to assist us to get weary and hostile after dismissing you. I get the

notion that others say, "There comes, everyone was biased, neglecting, since if he gets into a dilemma, he could abuse others." That, I believe, has not been favourable for me.

### Discrimination

Respondents stated that counselling them was as if they cannot cope except in rare cases. On the other hand, most of the children thought these unfavourable presumptions were connected to discriminatory actions.

> Even if you are OK, there is no confidence. If officials will come to meet with the counsellor to inquire about a problem, I would prefer not to meet anyone and want to be silent and stay alone. They don't think I'm real, which makes me feel awful. Because of it, they may share my details with other board members. What a coincidence I had a problem; later on, it may create more problems for my carrier.

**Fear of unforeseen repercussions and trust deficient**

"While I was in OH, few police officers never stopped saying I was criminal after all others said: 'me that he is a criminal, he is the bad guy."

A few individuals shared their experiences obtaining mental healthcare and the advantages they received. A lack of awareness about the kinds of support accessible to them and how to access those appeared to be an obstacle to receiving help, as did fear of being negatively judged and for lack of trust.

### Theme 9: Sleeplessness and aggressive behaviour

Most indicated they could not cope with stress and anxiety, although they reported sleeping problems. Some of them narrated they repeated the offence. "We fear the arrest or JJB will send back to the OH. So, we have a fear, and we cannot sleep."

"[In the OH] all children were in one room. It was just like a gaol, although we couldn't sleep."

Respondents stated they had less stress since returning from OH because they were with their families, and few of them got admission to the school. Most of them said they had difficulties with nightmares:

> Unpleasant dreams had occurred due to anxiousness, although they occurred more when I initially returned home. Yet the overwhelming sense of anxiety kept me awake at night, and sometimes it only did so when I was reminded of what I had done. Hardly a day goes by that I don't think about it or particular events that took place there, thus.
> We fear arrest or the Juvenile Justice Board will send them back to the observation home. So, we have a fear, and we can't sleep. With that, in the observation home, we don't have any services related to games, and there is only one room; all children are in one room. It was just like a gaol.

### Theme 10: Awareness level (outreach)
Sub-theme: Individual engagement sessions. Based on very few participants, one-on-one engagement sessions to initiate therapy would benefit those returning from OH. According to respondents.

"It will be preferable if the counsellor arranges individual counselling. One will feel more ease to discuss difficulties, which will assist with solving problems."

"Yet, it could be challenging to get us to attend supportive relationships since we might claim that we don't believe. So,

additional initiatives must be undertaken to let know that counsellors are available."

Others narrated that there is a lack of awareness regarding MHS and that if someone goes for MHS, he will be considered mentally ill. One child narrated as such:

"When anyone went for help related to mental health, most of the neighbours considered them *pagal* (mad). People will hesitate to talk; everyone will see with doubt and fear that he/she can harm us."

Society as well as officials must have awareness related to mental disorders and MHS. Then it will be easy for teenagers to avail of MHS comfortably.

Once upon a time, I shared everything with a counsellor and all other officials related to my aberrant behaviour. After a few days, some other offices came to me and asked about my behaviour, so how can we trust them? We can only trust over parents and friends; at least we will be safe.

Sub-theme: Community- and official-level engagements. It was found that most respondents needed to be made aware of the MHS.

"Such services must be because they can help us understand the difference between reality and our actions."

Still, with that, one of the respondents narrated: "Is ka kush nhi bn piye, is say dar lagta ha, jha pagal ha, is ka kush nhi ban sakta. Ise liye ma ghar say bhair nhi niklta hospital b nhi jata hou."

"Pagal" word is ubiquitous in Indian societies for those with mental illnesses or psychological problems (Murthy et al., 2020).

### Theme 11: Comfortable with MHS
Most participants agreed that having one-on-one meetings to begin services would help them feel more at ease with MHS. One said, for instance:

> It all comes down to the fact that I dislike psychiatry. I'm not sure I really get how it is with them. Someone telling me how I feel irritates me. I, too, find that group thing annoying. I don't particularly appreciate talking to huge gatherings of people. I like speaking with a select group rather than many individuals.

However, respondents felt uncomfortable with group therapy but were more comfortable with individual therapy.

### Theme 12: Confidentiality
Sub-theme: Fear of privacy breach and breakdown of confidentiality.

> More immediate access, yes. You don't need to tell your mates that you're simply going in to talk to someone. It would be useful. In my opinion, many inmates know that it should be kept private, but because you are still being watched, it is still there. I believe that this is a mental barrier that prevents them from speaking to someone.

Another respondent: "Well, I guess you would have to be with us to understand all this. Hard to talk with somebody that's never been there."

According to one respondent:

> I'm not sure. It doesn't appear to be bad. Our correctional home inmates dealt with things in a rather transparent manner. Just be real and sincere. One such of friend murdered his girlfriend after suffering a total mental. We won't hold him responsible. He's not to blame. Another issue would arise if you claimed to have been traumatised while not seeing anything. No one should encounter any issues as long as you are being true and honest. The only thing I can think of is that.

Others believed that everything would be OK if one were honest and trustworthy. Most of the respondents expressed concern that confidentiality would not be maintained. Delinquents feared that confidentiality would be violated.

"So, the staff and counsellor are utterly unfathomable to me."

"As every aspect of OH is different here. It seems that OH differs for each individual. I don't believe that if I have a problem, just my parent or another person can assist me."

As compared to service providers, children demonstrated more confidence in parents. A deeper-seated mistrust of health MHS is complicating things further.

"I don't trust this counsellor. Thus, I need to be more comfortable."

"Even though I do not believe others, I retained my difficulties secret from myself and won't disclose them to anybody else."

Respondents talked about privacy breach and confidentiality. Trust presents still another obstacle to appropriate service participation. Distrust may cause a person to feel unsafe. These juvenile delinquents had a severe problem with trust.

> The most crucial factor is trust. Thus, I would recommend counselling since I really valued their assistance and, honestly, really needed help at the time because I was in such a mess. Keeping things private would never benefit you.

As many children having trouble with the law reported they did not like attending their sessions and that the therapies were ineffective; it is probably not unexpected that the overall perception of MHS was negative.

"I like a few moments to sit while counsellors perform counselling because I don't enjoy disturbance. With that, one needs to attend individual sessions to develop trust."

"I lost my father, I had many issues, and I talked about many of them with staff members, but they didn't assist me."

Dealing with these individuals requires caution since their attitudes towards MHS could deteriorate if they believe their issues are getting worse. Some juveniles had unpleasant reactions to what is being done, including an in-depth evaluation and description of the problems. This defines the psychiatric profession and calls for much scrutiny. Despite the participants' general pessimism about using services and reluctance to go, some saw benefits. These children have also discussed trust and stigma.

## Discussion

Child delinquency, family income and educational level are directly correlated (Rekker et al., 2015; Garbarino and Plantz, 2017; Lochner, 2020). The researcher believed the participants in the research would provide comments revealing that the use of MHS would be correlated with stigma, as suggested by a review of literature. The thematic review showed that out of a total of 15 respondents, most believed that MHS was stigmatised (Tully et al., 2019). The remaining expressed concern that receiving MHS after leaving the OH would either prevent them from being hired or be problematic for getting a job. They may face problems getting admission to the school (Evans et al., 2018; Townsend et al., 2019). These findings somewhat corroborate the notion that youngsters who run afoul of the law experience stigmatisation when they seek mental healthcare after leaving the OH (Milin et al., 2016; Zola et al., 2017). Studies have found that OH does not have enough MHS (Underwood and Washington, 2016; Paul and Khan, 2019). However, it was interesting that most aberrant children interviewed claimed no stigma associated with obtaining MHS. But most minors claimed they had

not sought treatment despite exhibiting symptoms (Thompson et al., 2016) and various psychological symptoms (Firmin et al., 2016). The lack of seeking treatment supports the hypothesis that a stigma is attached (Stringer and Baker, 2018; Lievesley et al., 2020). Even so, delinquents found it easier to talk about their difficulties with associates (Rafedzi et al., 2016). It was determined that most individuals did not experience any official pressure to be less than candid about their mental health issues. The children's responses to other questions, however, were intriguing since they suggested there might be unspoken pressure from officials to refrain from seeking counselling or other required interventions (Ricciardelli et al., 2020). A few aberrant teenagers claimed they were engaged in several illegal actions and would be wrong if they told anyone about their time in the OH and, with that, their involvement in several illicit pursuits. This is intriguing since the younger hinted that if people judged him as a bad guy, he would not develop into a strong man and could not accomplish anything in the future. Another teenager claimed he did not want to use MHS because the officials always view with prejudice, interfering with their schooling (Snyder, 2015) and possibly their careers. If children stopped getting mental healthcare while in an observation home, their chances of finding employment might decrease (Chan, 2019). The stigma might also intensify (Moore et al., 2016).

The final hypothesis was that aberrant teenagers lack trust in counsellors. It was hypothesised that children feared there would be breaches in confidentiality because the counsellor who was coming may share everything with the JJB or other officials (Gibbs et al., 2015). The study's finding matches this hypothesis as the teenagers failed to go for the services, expressed a fear of being noticed by the various officials of the observation home, and many shared that they would feel more agreeable with their family and friends. Many aberrant children said they would share their problems with family members (Kapetanovic et al., 2019), but it has been seen that they are comfortable talking with friends (Lee et al., 2021). However, following a comprehensive examination of the data, it appears there is a bad stigma linked to obtaining psychological help, such as being labelled as a wrong person. Evidence did not explicitly support each assumption. It was also clear that youngsters who broke the law feared consequences, such as losing their jobs and having a counsellor with the JJB violate their privacy (Todres, 2022; Math et al., 2019; Birchley et al., 2017). However, the stigma, unexpected repercussions and privacy intrusions were not explicitly addressed by these youngsters.

## Role of social workers

Juvenile offenders often experience remorse and humiliation due to "their different delinquent behaviours" (Stuewig and McCloskey, 2005; Hoffman and Duschinski, 2020). These emotions may significantly discomfort individuals and exacerbate mental health conditions in that age category. Understanding the psychological effects of obtaining mental health treatment may inspire reforms to serve this demographic better. It is crucial to realise that stresses in correctional facilities have been demonstrated to impact psychological health while inmates are away from home and after they return (Fazel et al., 2008; Tasca et al., 2014). To effectively treat minors with MHS, it is vital to understand their particular requirements. This research offers three separate contributions to the field of social work. While dealing with children who have had a challenging reintegration, "psychiatrists, physicians, or nurses need to be aware of the delicate nature of reintegration" at the micro level (Baglivio

et al., 2017). This study investigated children's attitudes and beliefs in understanding what must be changed to improve mental health treatment. Such children may benefit from MHS and social work assistance, and the aberrant children see these services as beneficial (Whitney and Peterson, 2019). This study will provide social workers insights into constructively interacting with the target population and developing a strong therapeutic bond. According to studies, abnormal teenagers prefer to talk about their issues and activities, and processing these experiences may be beneficial (Hartwig and Myers, 2003; Vyas, 2016). This research will assist social workers in understanding MHS-seeking hurdles and facilitating transformation. As a result, services may be delivered successfully. Understanding the proper services is crucial since "the prevalence of mental health disorders connected to stress and adaptation indicates a need to build community-based service resources to manage and assist the afflicted persons" (Wells et al., 2004; Sellers, 2015; Rijo et al., 2016). Social workers can promote change due to this study to ensure that these children's needs are adequately met. Taking care of youngsters' emotional factors at OH will result in fewer stressed people who will perform better in the future (Abrams, 2013; Covington and Bloom, 2014). Through community-based programmes and activities, children participate in many forms of delinquency; social workers may restrict such activities (Davidson & Wolfred, 1997; Phillips, 1997; Henggeler, 2016). This research has generated information at the policy level that would enable efficient methods for identifying mental health issues and developing interventions for the targeted group. Limited studies are currently available to frame policy about effectively encouraging access to and delivering mental health treatment for such youngsters (Hassan, 2021). Last but not least, this research was necessary because social workers need to figure out how to connect with adolescents who have encountered psychiatric problems (Goldkind, 2011).

### Directions for research, policy and practice in social work

To ensure that all juvenile offenders get the care they need, the study was designed to catalyse further investigation. Further investigation is required to determine what obstacles prevent such adolescents from obtaining MHS and what initiatives might benefit counselling those who engage in delinquency. This research made the social work field aware of the stigma these minors face when they seek treatment, whether overtly acknowledged or not. It assisted in identifying the obstacles individuals confront while seeking assistance.

### Law enforcement agencies and NGOs

To connect delinquents with different skill development programmes, authorities may establish neighbourhood facilities for leisure activities, such as playgrounds and indoor games. To increase awareness and to conduct different events, the police and Community Based Organisations may work together (Dar and Mir, 2013; Maqbool and Khan, 2020). NGOs may bridge the gap between service providers and those who need assistance, support research capacity building and increase outreach to the community (Rather and Margoob, 2006).

### Family and educational institutions and other allied institutions

Schools may play a crucial role in preventative and intervention initiatives by enlisting the help of experts, such as social workers, psychiatrists, counsellors and physicians (Waxman et al., 1999; van Os et al., 2019), and multidisciplinary teams to create strategies for teaching children and parents about the value of family ties in prohibiting the development of delinquents. The professional team might also make a strategic plan (including community outreach, counselling, leadership training and team building) to lessen the trouble of some behaviours and support the child's improvement (Smith et al., 2022).

### Psychoeducation

Psychoeducational therapies are those in which people with mental problems are provided knowledge. Psychoeducation might assist individuals in comprehending their issues, talking about them with others and adopting coping skills afterwards. The study's main findings focused on stigma, help-seeking attitudes, help-seeking intentions and mental health literacy (Malla et al., 2019; Pandya et al., 2020). On the other hand, it can play an essential role in Kashmir as the delinquents suffer from various psychological problems (Carpenter and Sugrue, 1984). Psychoeducation was started in Kashmir in the 2000s through radio. As it was for the general population, the community responded excellently (Hamdani, 2003). Individuals have found that sharing problems with others, using muscular relaxation exercises, fostering healthy family interaction and engaging themselves with activities were advantageous. Several persons said they found it highly beneficial to convey the ideas surrounding the myths and realities of psychological health. After seeing the broadcast, everyone in their family felt obliged to discuss their issues (Hamdani, 2003). In the context of Kashmiri youth, information could be enhanced through psychoeducation, and comprehension would enable individuals with mental illnesses to cope with disorders better, improving prospects. Virtual psychoeducation has been more prevalent recently to improve access to resources like lectures and online modules (Kim et al., 2020; Migoya-Borja et al., 2020).

### Suggestions

1. MHS in Kashmir must be developed with community participation, awareness and mental health rehabilitation services.
2. For those living with persistent trauma, proper counselling services must be accessible.
3. Academicians, doctors and decision-makers must also develop strategies and regulations considering the particular demographic and mental health.
4. We advocate the creation of a comprehensive virtual MHS network for mental health issues. This strategy will increase the availability of services.
5. We should ensure availability of affordable MHS with prompt diagnosis and enhanced therapeutic follow-up.

**Open peer review.** To view the open peer review materials for this article, please visit http://doi.org/10.1017/gmh.2023.70.

**Data availability statement.** Not applicable due to the anonymisation of interview data.

**Author contribution.** M.M.A. led the search and screening of articles, interviews with respondents and data analysis. He drafted the initial and final manuscript and resolved inclusion criteria and data analysis; J.J. was integrally involved in theorising and research and contributed to the final paper.

**Financial support.** Central University of Kerala.

**Competing interest.** The authors declare none.

**Ethical consideration.** Before the interview, each participant completed an informed consent form after receiving a brief explanation of the study. The fact that participation was voluntary and that leaving at any time without consequence was made clear to participants. By coding the information from each participant, anonymity was protected. Most respondents were afraid because of the conflict in J&K; thus, data were carefully analysed as per their request. The method of data analysis chosen involved the researcher selecting coding themes from each question on behalf of the item and carefully examining the narratives and statements of respondents.

**Consent for publication.** Permission was obtained from each participant before the interview.

**Limitations.** The study's limitations include its limited sample size and focus on children from specific neighbourhoods. Delinquent children made up the sample. Hence it was impossible to extrapolate the results of this research to the overall youth population.

## Abbreviations

| | |
|---|---|
| APDP | Association of Parents of Disappeared Persons |
| APL | above poverty line |
| BPL | below poverty line |
| ICPS | Integrated Child Protection Scheme (Mission Vatsalya) |
| J&K | Jammu and Kashmir |
| JJ Act | Juvenile Justice Act |
| JJB | Juvenile Justice Board |
| JKCCS | Jammu Kashmir Coalition of Civil Society |
| MHS | mental health services |
| NGOs | non-governmental organisations |
| OH | observation home |
| PTSD | posttraumatic stress disorder |
| UNCRC | The United Nations Convention on the Rights of the Child |

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
