## [Reviewer Report]

Mohd Manshoor Ahmed

PhD. Research Scholar 

Department of Social Work 

Central University of Kerala,

Tejaswini Hills,Periye (PO),

Kasaragod (DT), Kerala-671320

INDIA

Email: ahmed.ssw072106@cukerala.ac.in

Date: 05-01-2k23

Respected Editor, 

We wish to submit an original research article entitled “Perceptions of Mental Health Services Among the Children Who are in Conflict with the Law in Jammu And Kashmir.” for consideration by Global Mental Health. 

We confirm that this work is original and has not been published elsewhere, nor is it currently under consideration for publication elsewhere.

This paper finds that Jammu and Kashmir is a conflict zone, and generations over the decades have suffered. However, teenagers are also part of a particular society and are primary victims of a specific conflict, and part of the conflict also. Due to that, they adopt aberrant behaviour. After arrest, adolescents are shifted to correctional homes for rehabilitation. This is significant because there is much need to generate awareness among the children who conflict with the law regarding mental health services to eradicate stigma, mistrust, and discrimination. With that, authorities must have to do more also. 

We believe this manuscript is appropriate for publication in the Global Mental Health. We have no conflicts of interest to disclose. 

We appreciate your consideration of this manuscript. 

Sincerely,

Mohd Manshoor Ahmed Dr Jilly John 

PhD. Research Scholar Assistant Professor

Department of Social Work Department of Social Work

Central University of Kerala Central University of Kerala

Email: ahmed.ssw072106@cukerala.ac.in Email: jillyjohn@cukerala.ac.in

---

## [Reviewer Report]

The article has been written well on an issue of particular importance to child protection. However, the article may be strengthened on the methodology part, the problem under study also needs to be strongly dealt within the existing system of child protection, while also highlighting the strength and weakness of the systems in place. The grounds for self reported perception also need to be thoroughly analysed. The article may be considered after the revision.

---

## [Reviewer Report]

Study lacks the conceptual frame and the whole enquiry was based on mere 15 respondents which later projected as quantifiable data elements. This whole situation lead to inconclusive projections which was made in the conclusion. Authors are suggested to revise the manuscript while focusing on qualitative methods of data analysis.

---

## [Reviewer Report]

Dear Author(s),

Thank you for the manuscript- The subject of the manuscript is very important, and there is no doubt that the Mental Health Services for the children in Jammu and Kashmir who are in conflict with the law are very important to consider because of the increasing trend in the development of mental disorders as a result of a conflict situation and various other factors like natural disasters.

The manuscript examined the Perceptions of Mental Health Services Among Children Who conflict with the Law in Jammu and Kashmir.

I have a few comments for your consideration:

1. Author(s) need to provide the full form of all the acronyms when used the first time, such as J&K, etc.

2. Limitations should not be provided in the abstract; they should be included in the limitations section, followed by a discussion.

3. Jammu and Kashmir is a conflict-affect zone where many conflicts related events occur where common people suffer, including children. Therefore, the question arises of why children break the law or do anti-social activities. Moreover, there is a need to provide an operational definition of the anti-social activities because the conceptualization of anti-social activities varies from country to country and from region to region within the same country. And it is also essential to understand whether the children consider these activities anti-social or not.

4. Genesis of the Kashmir conflict needs to be provided, touching the partition of the Indian subcontinent through the insurgency in the late 1980s to date. Literature related to the toll of the conflict in Kashmir needs to be provided in terms of deaths and injuries; torture and abuses (physical, emotional, verbal, sexual); physical, mental, and social health; and others (provide statistics).

5. Author(s) need to provide context regarding the number and nature of laws prevailing in the region and children’s life to understand their vulnerability.

6. There is the repetition of the sentences in the abstract, introduction, and other sections of the manuscript, which makes it monotonous and needs to be removed. There are typos and language issues throughout the manuscript, which need to be addressed.

7. Author(s) need to use the words criminal/ crime carefully because the age of the juveniles starts from 10 years.

8. The introduction has less important, lee relevant information, and some of the paragraphs seem repetitive, and it makes the overall structure of the introduction less clear. I would consider revising so that the distinction between sections is clearer. There is no theoretical framework provided, so the sub-heading ‘The Environment of Children’ should be provided through ecological modeling covering different layers and their exposures; accordingly, relevant literature needs to be provided to develop the theoretical framework. Author(s) need to provide the relevant literature in the sections on mental health services, stigma, and trust by combining them into one since they have relevance. The last paragraph should provide the study’s rationale, research questions, and objectives.

9. The methods section is very weak and less clear. There is ambiguous information provided regarding the study design, sample, sampling techniques, procedure, and description of tools. There is no information listed about the ethical aspects and analysis carried out. Author(s) need to work more to standardize the complete methods section. The methods section should be written sub-section-wise to make it more clearer.

10. The sample is very small, and there is concern about the context of the settings from which participants have been taken; that is, eight children have been selected from the Kashmir region and seven from the Jammu region, but the two regions differ in terms of the prevalence of conflict-related events. The Kashmir region is significantly impacted by armed conflict, whereas in the Jammu region, conflict-related events seldom take place. Please justify.

11. The author(s) have not provided the background information of the sample, which is pivotal to understand the family dynamics and various other indicators. The information about involvement in anti-social activities and the health status of the parents and other family members is also not provided, which could help understand any family history.

12. Details related to the analysis of the results have not been provided. It needs to be made clear how the data has been analysed. Results have been randomly provided, making it difficult to understand what the findings speak. The author(s) needs to work more on the analysis part so that results can be interpreted meaningfully.

13. Findings have not been properly discussed by citing the relevant literature, which makes discussion week. The important sections of limitations and recommendations/ suggestions have not been listed, which could make the findings meaningful and manuscript strong.

14. Also, areas lacking attention and future suggestions for the researchers have not been provided. There is no mention of the role of health professionals; clinicians; law enforcement agencies, social workers, policymakers, educational institutions, and family. The importance of law and psychoeducation in terms of awareness needs to be included.

I think this is an interesting manuscript, but it needs more clarification and information completion with respect to all the sections, which makes it underwhelming.

Thank you, and good luck.

---

## [Reviewer Report]

Revision Cover Letter 

Address to the Editor Date: 06/04/2023

Dr Victoria Lane

Editorial Office, 

Cambridge Prisms: Global Mental Health

Dear Editor, Global Mental Health 

We want to thank you for the letter dated 28/02/2k23 and the opportunity to resubmit a revised copy of this manuscript. We also express our gratitude to the reviewers for their encouraging words and insightful suggestions for improvement.

We have resulted in an improved revised manuscript titled “Perceptions of Mental Health Services Among the Children Who are in Conflict with the Law in Jammu and Kashmir”, which you will find uploaded alongside this document. The manuscript has been changed to address the reviewer’s comments, those addressed and uploaded in the response session. 

We really hope that the revised manuscript will be approved for publication in the concerned journal.

Sincerely yours,

Mohd Manshoor Ahmed Dr Jilly John 

PhD. Research Scholar Assistant Professor

Department of Social Work Department of Social Work

Central University of Kerala Central University of Kerala

Email: ahmed.ssw072106@cukerala.ac.in Email: jillyjohn@cukerala.ac.in

---

## [Reviewer Report]

1. The study is quantitative but the sample size is too small, no where the size of the population is mentioned.

2. Sampling frame also needs to be added

3. The section in which themes are mentioned is not bearing any title, which is making the content ambiguously left for reader’s inference.

4. Grammatical errors are still present, sentences are also left incomplete.

---

## [Reviewer Report]

• On many occasions author made assumptions which are claims without empirical evidences, e.g. children in conflict with law make mob,

• Initial background for Jammu and Kashmir as a disputed territory reflect certain bias from the author’s nationality sentiments

• There is a lack of clear evidence within the paper, which establish the causality and degree of relationship among exposure to violence lead by defence forces and mental disorder among the civilians who became victim of the same.

• Qadir (2021) mentioned on many occasions but not mentioned in the reference list.

• Post abrogation of Article 370 various legal mechanisms evolved in the UT of J&K and the same has not been acknowledged in the paper. Paper is still projecting legal and administrative scenario previous to that abrogation.

• Conceptual Frame and objectives of the research make the context of conflict irrelevant.

• Thematic analysis of the study demands comprehensive thematic classification involving critical analysis, rather limiting majorly to narratives.

• Many sentences were not complete and there is a serious need for proofing of the whole paper.

---

## [Reviewer Report]

Dear author,

Thanks for submitting a revised version of your article, along with your responses to the review comments. While some aspects have improved, both reviewers, and myself as editor still noticed significant concerns, that make the paper not suitable for publication at this point. 

First of all, you added a historical overview of the J&K region, which is too detailed in the light of the aim of your study. We suggest you to greatly shorten this paragraph.

Further, as also noted by the reviewers, there are multiple errors throughout the paper, such as typos and and unfinished sentences. A proofread by a native English speaker needs to be carried out.

Another problem that needs to be addressed is that there are still statements throughout that lack references to the scientific evidence.

Finally, was the study evaluated by an ethics committee? This should be mentioned.

---

## [Reviewer Report]

Mohd Manshoor Ahmed

PhD. Research Scholar 

Department of Social Work 

Central University of Kerala,

Tejaswini Hills,Periye (PO),

Kasaragod (DT), Kerala-671320

INDIA

Email: ahmed.ssw072106@cukerala.ac.in

Date: 05-10-2k23

Respected Editor, 

We wish to submit an original research article entitled “Perceptions of Mental Health Services Among the Children Who are in Conflict with the Law in Jammu And Kashmir.” for consideration by Global Mental Health. 

We confirm that this work is original and has not been published elsewhere, nor is it currently under consideration for publication elsewhere.

This paper finds that Jammu and Kashmir is a conflict zone, and generations over the decades have suffered. However, teenagers are also part of a particular society and are primary victims of a specific conflict, and part of the conflict also. Due to that, they adopt aberrant behaviour. After arrest, adolescents are shifted to correctional homes for rehabilitation. This is significant because there is much need to generate awareness among the children who conflict with the law regarding mental health services to eradicate stigma, mistrust, and discrimination. With that, authorities must have to do more also. 

This manuscript is appropriate for publication in the Global Mental Health. We have no conflicts of interest to disclose. 

We appreciate your consideration of this manuscript. 

Sincerely,

Mohd Manshoor Ahmed Dr Jilly John 

PhD. Research Scholar Assistant Professor

Department of Social Work Department of Social Work

Central University of Kerala Central University of Kerala

Email: ahmed.ssw072106@cukerala.ac.in Email: jillyjohn@cukerala.ac.in